# Tubulin-Dependent Transport of Connexin-36 Potentiates the Size and Strength of Electrical Synapses

**DOI:** 10.3390/cells8101146

**Published:** 2019-09-25

**Authors:** Cherie A. Brown, Cristiane del Corsso, Christiane Zoidl, Logan W. Donaldson, David C. Spray, Georg Zoidl

**Affiliations:** 1Department of Biology, York University, Toronto, ON M3J 1P3, Canada; cbrown88@yorku.ca (C.A.B.); czoidl@yorku.ca (C.Z.); logand@yorku.ca (L.W.D.); 2Department of Biophysics and Physiology, Federal University of Rio de Janeiro-RJ, 21941-901 Rio de Janeiro, Brazil; cdcorsso@biof.ufrj.br; 3Department of Neuroscience, Albert Einstein College, Bronx, NY 10461, USA; david.spray@einstein.yu.edu; 4Department of Medicine, Albert Einstein College, Bronx, NY 10461, USA; 5Department of Psychology, York University, Toronto, ON M3J 1P3, Canada

**Keywords:** Connexin-36 (Cx36), gap junction, electrical plasticity, tubulin and microtubules, cytoskeleton, transport

## Abstract

Connexin-36 (Cx36) electrical synapses strengthen transmission in a calcium/calmodulin (CaM)/calmodulin-dependent kinase II (CaMKII)-dependent manner similar to a mechanism whereby the N-methyl-D-aspartate (NMDA) receptor subunit NR2B facilitates chemical transmission. Since NR2B–microtubule interactions recruit receptors to the cell membrane during plasticity, we hypothesized an analogous modality for Cx36. We determined that Cx36 binding to tubulin at the carboxy-terminal domain was distinct from Cx43 and NR2B by binding a motif overlapping with the CaM and CaMKII binding motifs. Dual patch-clamp recordings demonstrated that pharmacological interference of the cytoskeleton and deleting the binding motif at the Cx36 carboxyl-terminal (CT) reversibly abolished Cx36 plasticity. Mechanistic details of trafficking to the gap-junction plaque (GJP) were probed pharmacologically and through mutational analysis, all of which affected GJP size and formation between cell pairs. Lys279, Ile280, and Lys281 positions were particularly critical. This study demonstrates that tubulin-dependent transport of Cx36 potentiates synaptic strength by delivering channels to GJPs, reinforcing the role of protein transport at chemical and electrical synapses to fine-tune communication between neurons.

## 1. Introduction

Synaptic plasticity at glutamatergic chemical synapses involves highly orchestrated molecular and morphological processes that fine-tune communication between neurons. Short- to long-term functional and morphological adaptations require dynamic trafficking and turnover of channel and receptor proteins. While the molecular machinery involved in chemical synaptic plasticity and the respective signaling network has been extensively investigated, fewer details are known about the plasticity of electrical synapses. In vivo, modifiable responses of electrical synapses have been found in both lower and higher vertebrates [1,2]. In vitro, a phenomenon coined “run-up” was found to rapidly and profoundly strengthen gap-junctional conductance exclusively at Cx36 gap-junction plaques (GJPs); an observation later was shown to occur in a Ca^2+^/calmodulin-dependent kinase II (CaMKII)-dependent manner [3,4,5]. The Cx36 plasticity phenomenon parallels the interaction of calmodulin (CaM)/CaMKII with the NR2B subunit of the N-methyl-D-aspartate (NMDA) receptor, in which phosphorylation of NR2B regulates the transient influx of Ca^2+^, a critical mediating step in synaptic plasticity of chemical synapses. 

The regulation of synaptic strength through ion channels, NMDA and α-amino-3-hydroxy-5-methyl-4-isoxazolepropionic acid (AMPA) receptor recruitment, and turnover is potentiated by calcium entry through NMDA receptors [6,7,8,9,10,11,12]. The microtubule network together with the axon cytoskeleton plays a crucial role in this process due to its ability to deliver cargo quickly and selectively to synaptic terminals [13,14,15]. NMDA receptor subunit NR2B requires a dynamic microtubule network for insertion into the synaptic contact site, and this is accomplished by direct interactions of the NR2B carboxyl-terminal (CT) domain with tubulin [16,17,18]. The Cx36 interaction with tubulin was found initially in a proteomic screen [19], leading us to further determine the mechanisms by which Cx36 interacts with tubulin and whether this interaction with the microtubule network contributes to the “run-up” plasticity seen at Cx36 gap junctions. 

Using the mouse Neuro-2a neuroblastoma cell line as a platform for exogenous expression of wild-type and mutant Cx36 proteins, we demonstrate that Cx36 binds tubulin through a conserved binding motif, which is distinct from the confirmed motif of Cx43 [20,21]. The Cx36-tubulin binding motif possesses a unique overlap with the binding sites for CaM and the CaM/CAMKII complex [4,20,22], a feature that is exclusive to the Cx36 isoform and its orthologs. Paired cell patch-clamp electrophysiology substantiated the role of tubulin binding in electrical synaptic plasticity. Site-directed mutagenesis delineated positional consequences of the tubulin-binding motif, and pharmacological interventions showed that the dynamic vesicular transport of Cx36 to the cell membrane and subsequent formation of gap-junction plaques was facilitated by the interaction of tubulin with Cx36. Since the NR2B subunit of the NMDA receptor binds tubulin and CaM/CaMKII at nonoverlapping binding sites, we conclude that Cx36 interacts with similar molecular machinery seen at chemical synapses but has adopted a unique mode of regulating synaptic plasticity through the presence of signature binding domains. This adaptation suggests a specialization required for on-demand transport of Cx36 specifically into the distinct presynaptic and postsynaptic compartments of contact sites where electrical synapses can be found in mature neurons of the vertebrate nervous system [23,24]. 

## 2. Materials and Methods

### 2.1. Plasmid Constructs

*Rattus norvegicus* Cx36 (Accession number: NP_062154.1) was used in this study. Wild-type Cx36 tagged with enhanced green fluorescent protein (EGFP) and the carboxyl-terminal mutant Cx36Δ279–292-EGFP (denoting the deletion of amino acids 279–292) expression plasmids were generated previously [3,5]. Cx36Δ279–292 was isolated and subcloned into the EGFP expression vector pEGFP-N1 (Clontech, Mountain View, CA, USA). EGFP expression plasmids for wild-type Cx36 and Cx36Δ279–292 were used for the subcloning of Cx36 sequences into pcDNA 3.1-MCS-BirA (R118G)-HA. The expression vector pcDNA 3.1-MCS-BirA (R118G)-HA (pcDNA3.1 BirA*) was purchased from Addgene (Cambridge, MA, USA). Oligonucleotides for single amino acid substitutions in the tubulin-binding region were designed using the NEBaseChanger tool (New England Biolabs Inc., Boston, MA, USA) and synthesized by Integrated DNA Technologies (IDT, Coralville, Iowa, USA). Alanine-scanning was performed using the Q5 Site-Directed Mutagenesis Kit as per the manufacturer’s protocol (New England Biolabs Inc., Boston, MA, USA). All plasmid constructs were screened to confirm the correct directionality by restriction endonuclease digest and sequence verified by Eurofins MWG Operon LLC (Huntsville, AL, USA). The pmCherry-alpha-tubulin (Addgene, 49149), pDsRed2-ER (Clontech, 6982-1), pDsRed-Monomer-Golgi Vector (Clontech, 632480), and the pDsRed-Monomer-Caveolin 1 (Clontech) vectors were purchased as described. 

### 2.2. Cell Line, Cell Culture, and Transient Transfection

Mouse neuroblastoma 2a (Neuro-2a) cells (ATCC^®^, CCL-131, Manassas, VA, USA) were cultured in high-glucose Dulbecco’s Modified Eagle Medium (DMEM) supplemented with 10% fetal bovine serum (FBS), 1% nonessential amino acids (NEAA), and 1% penicillin/streptomycin as described [3]. The cell line was previously authenticated in-house using a combination of morphological confirmation, differentiation capacity, and identifying neuronal markers via PCR. There was no evidence of mycoplasma contamination. All cell-culture reagents were purchased from Thermo Fisher Scientific (Burlington, ON, Canada) or Sigma-Aldrich (Oakville, ON, Canada). For transient transfections, Neuro-2a cells were grown to 60% confluency (approximately 0.3–1.92 × 10^6^) overnight in 19 mm (24 well), 100 mm, or 35 mm MatTek culture dishes. Cells were transfected using the Effectene Transfection Reagent Kit (Qiagen, Valencia, CA, USA) with 200 ng DNA, as per the manufacturer’s protocol, and all analyses were performed 48 h post-transfection.

### 2.3. Protein Isolation and Immunoblot Analysis

Transiently transfected Neuro-2a cells were washed once with divalent free (DF) phosphate-buffered saline (PBS) prior to lysis. Whole-cell protein lysates were prepared directly in 1× Laemmli sample buffer (50 mM Tris-HCl, pH 6.8, 2% sodium dodecyl sulfate (SDS), 3% β-mercaptoethanol) and heated for 3 min before gel separation. Twenty micrograms of proteins were fractionated by sodium dodecyl sulfate-gel electrophoresis (SDS-PAGE) on 5% stacking and 10% running gels and transferred onto a 0.2-µm nitrocellulose membrane using the Trans-Blot^®^ Turbo™ Transfer System (both Bio-Rad, Mississauga, ON, CA). Membranes were blocked with 1× iBind™ Solution (horseradish peroxidase (HRP) or alkaline phosphatase (AP) detection), and immunoblotting was carried out using the iBind Western System (Thermo Fisher Scientific, Burlington, ON, Canada) as per the manufacturer’s protocols. Primary antibodies and enzyme conjugates were diluted 1:10,000 (anti-β-actin: clone AC-15, A5441 and anti-HA: clone HA-7, H9658, Sigma-Aldrich, Oakville, ON, Canada), 1:1000 (anti-GFP: clone F56-6A1, sc-53882, Santa Cruz Biotechnology, Dallas, TX, USA; anti-β-tubulin: clone TUB 2.1, ascites fluid, T4026, Sigma-Aldrich, Oakville, ON, Canada), and 1:2500 (Streptavidin-HRP: 926-68079, Invitrogen, Burlington, ON, Canada). Secondary antibodies were diluted 1:10,000 (IRDye^®^800CW Goat Anti-Mouse: 926-32210, Li-Cor Biosciences, Lincoln, NE, USA) and 1:15,000 (IRDye^®^680LT Donkey Anti-Rabbit, Li-Cor Biosciences Lincoln, NE, USA). Fluorescent signals were detected using the Odyssey^®^ CLx Infrared Imaging System (Li-Cor Biosciences, Lincoln, NE, USA) as per the manufacturer’s protocol using default settings. 

### 2.4. In Vivo Affinity Capture of Biotinylated Tubulin Using BioID

On a 100-mm dish, (approximately 5.28 × 10^6^) transfected Neuro-2a cells expressing Cx36 wild-type or Cx36Δ279–292-BirA* fusion proteins were incubated with 50 µM biotin for 24 h as previously described [25]. Cells were washed three times with PBS (without magnesium and calcium) and lysed at 4 °C with NP-40 cell lysis buffer (Invitrogen, Camarillo, CA, USA) supplemented with 1× protease inhibitor cocktail (Thermo Scientific, Burlington, ON, Canada) and 5 units/mL benzonase (EMD Millipore, Etobicoke, ON, Canada). Approximately 400 µL of extracts were subjected to centrifugation at 13,000× *g* and incubated at 4 °C overnight with 180 µL of Dynabeads (MyOne Streptavidin C1, Invitrogen, Burlington, ON, Canada). Collection of the beads with protein candidates attached, washes, elution, and preparation for western blot or mass spectrometry analysis were performed as described [25]. 

### 2.5. Pharmacology

Tubulin-dependent trafficking of Cx36 to the gap-junction plaque was analyzed using Neuro-2a cells expressing EGFP-tagged wild-type Cx36 protein treated with either Colchicine (100 µM, Sigma-Aldrich) or Paclitaxel (20 µM, Sigma-Aldrich) to disrupt or stabilize microtubules, respectively. Neuro-2a cells were incubated with a drug at 37 °C, 5% CO_2_, for 10 min prior to use in Fluorescence Recovery After Photobleaching (FRAP) or Total Internal Reflection Fluorescence (TIRF) studies. 

### 2.6. Electrophysiology

Neuro-2a cell pairs expressing EGFP-tagged wild-type Cx36 or Cx36Δ279–292 were used for dual whole-cell patch clamp recording. In brief, Neuro-2a cells were perfused in a bathing solution supplemented with 100 µM Colchicine for 1, 3, or 24 h and the run-up was induced as previously described [5]. Whole-cell patch clamp recordings were performed as previously described [5,26] over a time course up to 70 min at room temperature. 

### 2.7. Fluorescence Microscopy and Image Processing 

#### 2.7.1. Fluorescent Imaging and Gap-Junction Plaque Measurements

Transfected Neuro-2a cells were fixed with 4% formalin for 15 min at room temperature. Cells were subsequently rinsed with PBS and distilled water prior to mounting with Fluoroshield™ with DAPI (Sigma, Oakville, ON, Canada) mounting medium. Mounted slides were visualized using a Zeiss LSM 700 confocal microscope controlled by the ZEN 2010 software program (Oberkochen, Germany) at room temperature. Cells were imaged using the Plan-Apochromat 63x/1.40 Oil DIC M27 or the EC Plan-Neofluar 40x/1.3 Oil M27 oil immersion lenses. Images were generated at a high resolution of up to 2048 × 2048 pixels and an average of 4 scanning repetitions in the single plane or each plane of a Z-stack. The single or multi-track mode was selected accordingly for the compilation of localization studies. The gap-junction plaque area (µm^2^) was measured using the ImageJ (NIH, Bethesda, MD, USA) free-hand tool. The percentage of Cx36 gap-junction-paired cells versus the total paired Cx36-expressing cells was measured as previously described [27]. Mander’s overlap coefficient was used to quantify the overlap between corresponding fluorescent signals derived from Cx36 and cell organelles in co-transfected cells. Mander’s overlap coefficient was calculated using the ZEN 2010 program. Co-localization was accepted at a Mander’s coefficient > 0.5; this threshold indicated that the occurrence of true co-localization surpassed the probability of chance. 

#### 2.7.2. Fluorescence Recovery After Photobleaching (FRAP)

For the quantitative analysis of connexon trafficking and gap-junction regeneration, FRAP was performed 48 h post-transfection using the Zeiss LSM 700 confocal microscopy in combination with Zeiss 63X (Plan-Apochromat, DIC M27 1.4) oil immersion lens and the Zen 2010 software. The microscope was equipped with an incubation chamber to maintain cells at 37 °C in DMEM without phenol red. Since previous reports [20,28] have described connexon replenishment occurring at the lateral ends of the gap-junction plaque, this was selected as the region of interest (ROI). ROIs were manually drawn at the non-junctional plasma membrane and lateral ends of the gap-junction plaque using the rectangle tool. ROIs were bleached using the 405-nm laser line at 70% emission strength with a single event of 20 iterations. Prior to recording, confocal images were optimized along the focal plane yielding in the largest GJP surface area. The mobile fraction (%) and intensity of fluorescence (RFU) were recorded in vivo for up to 200 s per ROI, and fluorescence values were collected at 1-s intervals. Observations were recorded with a pinhole of 1.0 airy units (AU) with a scan area of 31.1 µm × 31.1µm, zoom level between 1–3, and resolution of 512 × 512 pixels. Consistent parameters were used across all experiments. Experiments with considerable fluctuations in fluorescence intensity or ROI frame-shifts were not included in the analyses. Background subtraction was carried out with an ROI of the extracellular space to improve the signal-to-noise ratio. Loss of fluorescence was corrected next by expressing the previous values as a fraction of an unbleached control ROI. FRAP recordings were then normalized by expressing the corrected fluoresces as a fraction of the averaged pre-bleached values. The mobile fraction (M*_f_* = (F_max_ − F_0_)(1 − e^−*kt*^)) and half-time of recovery (T_1/2_ = ln 2/*k*) were calculated by the mono-exponential association equation in GraphPad: F*_t_* = F_0_ + (F_max_ − F_0_)(1 − e^−*kt*^), where F*_t_* is the fluorescence after background subtraction at time *t*; F_0_ and F_max_ are the fluorescence values immediately after and at the end of recovery following the bleaching event, respectively; and *k* is the first-order rate constant for recovery. Repopulation figures and X-Y scatterplots were done using the R software. Sample sizes ranged based on the efficiency of gap-junction plaque formation.

#### 2.7.3. Total Internal Reflection Fluorescence (TIRF).

For all experiments, Neuro-2a cells were plated onto 35-mm MatTek dishes and co-transfected with pShuttle-mCherry-Tubulin and the pEGFP-Cx36 wild-type or mutants as previously outlined. Time-lapsed TIRF microscopy was performed using the Zeiss Observer.Z1 spinning-disk microscope in combination with the Zeiss 100X (Plan-Apochromat, DIC, M27, 1.46) oil immersion lens, Photometrics Evolve™512 camera, and the Zen 2 (2014) software. The microscope was equipped with an incubation chamber to maintain the temperature at 37 °C and CO_2_ levels at 5%. Cells were imagined in DMEM without phenol red. Images were acquired at a resolution of 512 × 512 pixels in 20-s intervals for a total duration of 400 s. Images were processed in Imaris (Zurich, Switzerland) by tracking single particles expressing EGFP within the TIRF field. 

### 2.8. Molecular Modeling

A helical segment spanning amino acids 47–62 of the RB3 protein stathmin-like-domain X-ray structure [29] was substituted with a sequence from the Cx36 carboxy tail region (277–292) using PyMOL 1.8.6 and further refined with the FlexPepDock module of Rosetta 3.9 [30] 

### 2.9. Statistics

Beeswarm boxplots were generated in GraphPad Prism 6 (GraphPad Software, La Jolla, CA, USA). Plaque repopulation graphs and mobile fraction vs. half-time of recovery XY scatterplots were created in R-statistical program (The R Foundation, Aukland, New Zealand). All data are represented as the mean ± SEM. Unless otherwise stated, statistical analysis was carried out using SPSS (IBM Corporation; Armonk, NY, USA) and significance was determined by the Kruskal–Wallis test. For the FRAP data, statistical analysis was carried out using GraphPad and statistical significance was determined by the Kruskal–Wallis test followed by a Dunn’s multiple comparison test. *p* < 0.05 was considered significant. XY plots of the mobile fraction and half-time of recovery represented the mean ± CI. All comparisons were made to the wild-type (WT) control unless otherwise stated. Sample sizes and *p* values are provided throughout the manuscript and figure captions. 

## 3. Results

### 3.1. Cx36 Interaction with the Tubulin Cytoskeleton is Required for Electrical Plasticity

The interaction between tubulin at the carboxy-terminal (CT) of Cx36 was initially identified by mass spectrometry using both affinity-purified samples and immunoprecipitated protein complexes from murine brain lysates [19]. This result was independently confirmed by a BioID screen using a full-length rat Cx36 protein as bait in the Neuro-2a mouse neuroblastoma cell line (Figure 1A). Within the Cx36-CT, a potential tubulin site was identified by sequence comparison with the analogous tubulin binding sequence in connexin-43 (Cx43) [20,21]. This sequence overlapped with the previously reported binding sites for calmodulin (CaM) [22,31] and the calcium/calmodulin-dependent protein kinase II (CaMKII) [4], all of which demonstrated high conservation across Cx36 orthologs belonging to the major vertebrate Chordata subphylum (Figure 1B). Since the predicted tubulin-binding site and the established CaMKII/CaM-binding sites are situated within a fourteen amino-acid segment, any mechanistic model must consider competition among these proteins. This signature motif was absent in other connexin isoforms (Appendix A).

In previous studies [3,4,5], we have proposed that microtubule-dependent trafficking of Cx36 serves as a potential modality in achieving run-up plasticity by facilitating activity-dependent transport of Cx36 to gap-junction plaques. Here, the predicted role of Cx36 in run-up plasticity was demonstrated by dual whole-cell patch clamp experiments measuring electrical coupling dynamics in pairs of Neuro-2a cells expressing EGFP-tagged Cx36 constructs. Targeted ablation of the putative tubulin-binding motif [3,5] was used to investigate the involvement of this motif to run-up plasticity. In Neuro-2a cell pairs with a gap-junction plaque (GJP) present at the juxtamembrane, gap-junctional (GJ) conductance increased by a factor of about 10 during a ten-minute recording; by contrast, in cells expressing the Cx36Δ279–292 mutant, this run-up was severely blunted (Appendix A). This result showed that GJPs formed by Cx36 without the shared binding region for tubulin and CaM/CaMKII had reduced potential for plasticity. 

To determine whether a tubulin-specific component was involved in run-up plasticity, the microtubule network was manipulated by bath application with Colchicine, a selective agent which inhibits tubulin polymerization. The application of 100 µM Colchicine for either 1 h or overnight suppressed the junctional currents by approximately 70% (1 h) or 80% (24 h) compared to untreated controls (in nS, WT: 1.2 ± 0.1, *n* = 6; 1 h Colc: 0.2 ± 0.03, *n* = 5, *p* < 0.01 × 10^−2^; O/N Colc: 0.3 ± 0.02, *n* = 6, *p* < 0.01 × 10^−2^) (Figure 1C,D). An increase in GJ conductance was reversibly blocked when 100 µM Colchicine was applied for 20 min and washed out afterward (Appendix A). In a related experiment, monomeric tubulin (50 µM) was used as a competitive inhibitor to polymerized microtubules, which are known to be responsible for transport mechanisms. Here, intracellular perfusion with monomeric tubulin through the recording electrode was equally sufficient to block the time-dependent increase in junctional currents (in nS, WT: 1.5 ± 0.1, *n* = 4; tubulin: 0.2 ± 0.2, *n* = 5, *p* < 0.01 × 10^−2^) (Figure 1E,F). These results showed that the inhibition of tubulin polymerization abolished run-up plasticity.

### 3.2. Inhibition of the Cx36-Tubulin Interaction Promotes Intracellular Localization 

The involvement of the microtubule network in Cx36 delivery to the plasma membrane was studied by comparing the Cx36Δ279–292 mutant and wild-type control. Forty-eight hours post-transfection, no apparent expression differences were detected by immunoblot analysis (Figure 2A). Localization studies confirmed the Cx36 protein forming gap-junction plaques (GJPs) at the juxtamembrane (Figure 2B); however, Cx36Δ279–292-expressing cells showed frequent annular junctions (Figure 2C). Quantitatively, the mean area of Cx36Δ279–292 GJPs was significantly smaller than the WT control (WT: 2.2 ± 0.2 µm^2^, *n* = 28; Δ279–292: 1.6 ± 0.1 µm^2^, *n* = 26, *p* = 0.17 × 10^−1^) (Figure 2D) and less GJPs were formed between cell pairs expressing Cx36Δ279–292 (WT: 31.1 ± 3.8*%*, *n* = 30; Δ279–292: 8.7 ± 1.6*%*, *n* = 30, *p* < 0.01 × 10^−1^) (Figure 2E). We conclude that the Cx36Δ279–292 mutation interfered with transport to the cell membrane, accounting for the smaller and less frequent GJPs.

To determine the subcellular localization of the tubulin-binding deficient mutant, Cx36Δ279–292-EGFP or the Cx36 wild-type control were co-expressed with mCherry-tagged tubulin or the DsRed-monomeric protein tagged with (i) the ER retention signal KDEL, (ii) amino acids 1–60 of human galactosyltransferase, or (iii) caveolin-1. This approach permitted the visualization of the microtubule network, endoplasmic reticulum (ER), Golgi, and transport vesicles. Mander’s overlap coefficient (MOC) was used to quantify co-localization as the overlap between corresponding fluorescent signals contributed by Cx36 and cell organelles. Cx36Δ279–292 displayed less co-localization to the ER (WT: 0.69 ± 0.04, *n* = 30; Δ279–292: 0.57 ± 0.01, *n* = 25, *p* < 0.01 × 10^−1^) and Golgi organelles (WT: 0.65 ± 0.03, *n* = 30; Δ279–292: 0.46 ± 0.04, *n* = 15, *p* < 0.01 × 10^−1^), indicating that Cx36 vesicles were not primarily associated with the ER–Golgi complex. A marked reduction in co-localization to the tubulin cytoskeleton was observed for the Cx36 mutant expression in comparison to the wild-type protein (WT: 0.55 ± 0.02, *n* = 34; Δ279–292: 0.45 ± 0.02, *n* = 36, *p* = 0.20 × 10^−1^). Co-localization of Cx36Δ279–292 to caveolin-1 was indistinguishable from wild-type (WT: 0.59 ± 0.03, *n* = 40; Δ279–292: 0.58 ± 0.03, *n* = 33, *p* = 4.09 × 10^−1^) (Figure 2F and Appendix A); however, Cx36Δ279–292 intracellular vesicles were significantly larger (Vesicular Diameter, WT: 605 ± 39 nm; Δ279–292: 802 ± 50 nm, *n* = 50, *p* < 0.01 × 10^−1^) (Figure 2G). Our results suggest that tubulin mediates the transport of Cx36 starting at the ER. Whether the vesicular accumulation of Cx36Δ279–292 following release from the ER-Golgi complex represents the activation of the unfolded protein stress response or related cellular responses to misfolded or unfolded Cx36 mutant proteins was not investigated. 

The next goal was to elucidate the dynamics of trafficking of individual vesicles in vivo using TIRF microscopy (Figure 3A). Vesicles with Cx36Δ279–292-EGFP cargo were illuminated in the submembrane space by the evanescent wave known as the TIRF field. These vesicles were less mobile than wild-type controls, demonstrating impairments in both displacement length (WT: 8.7 ± 0.5 µm, *n* = 475; Δ279–292: 6.4 ± 0.5 µm, *n* = 304, *p* = 0.26 × 10^−1^), and mean speed (WT: 0.1 ± 0.01 µm/s; Δ279–92: 0.1 ± 0.01 µm/s, *p* < 0.01 × 10^−1^) (Figure 3B). We concluded that both temporal and spatial mechanisms of trafficking of Cx36 were impacted by the loss of the tubulin-binding region.

How were Cx36 connexons delivered to GJPs in the absence of tubulin binding? In general, connexons can replenish GJPs by two distinct pathways. Targeted direct delivery of connexons to the GJP relies on a complex of cytoskeletal and adherence-junction proteins [32]. Alternatively, connexons are routed to the plasma membrane and subsequently recruited into the GJP at the lateral ends [28,33,34]. This process occurs despite the ability of microtubules to anchor directly at the GJP [21]. 

To answer this question, fluorescence recovery after photobleaching (FRAP) was used to determine recovery kinetics of EGFP-tagged Cx36 wild-type or Δ279–292 proteins imaged at 1.0-s intervals pre- and post-bleach iterations. For visualization and quantification of connexon recovery, the fluorescence intensity of each region of interest (ROI) during the post-bleach recovery phase was corrected for background noise and normalized against the pre-bleached fluorescence intensity values. After bleaching, we observed that the plasma membrane (PM) of wild-type Cx36 recovered more efficiently than at the GJP (mobile fraction (M_f_) in %, GJP: 25.6 ± 0.6, *n* = 28; PM: 69.6 ± 3.6 *n* = 12, *p* < 0.01 × 10^−2^; half time of recovery (T_1/2_), GJP: 11.5s; PM: 23.5s) (Figure 4A,D). The higher M*_f_* seen at the PM suggested that, under control conditions, Cx36 is initially delivered to the plasma membrane and subsequently routed to the plaque, where it becomes stabilized, supporting the mechanisms previously described [28,34,35]. In Cx36Δ279–292-expressing cells, the M_f_ was again higher at the PM than at the GJP (GJP: 50 ± 0.5%, *n* = 22; PM: 97.1 ± 1.9%, *n* = 10, *p* < 0.01 × 10^−2^), suggesting that no changes were made to the route of delivery. Cx36Δ279–292 expression also led to a T_1/2_ that was faster at the PM and GJP (GJP: 0.9s; PM: 7.3s) than the WT (Figure 4B,D). We conclude that the interaction with the tubulin cytoskeleton aided in the temporal aspects of Cx36 incorporation from the PM into the GJP in addition to quantity of delivery, which was verified by GJP size and formation. 

In-depth comparisons of FRAP at the GJP revealed that the mutant GJPs were significantly more mobile (*p* < 0.01 × 10^−2^). This loss of stability was also accompanied by a large reduction in the T_1/2_, reflecting dynamics that were consistent with diffusion outcomes (Figure 4C,D). Overall, the results advocate for a mechanism in which tubulin interactions aid in the temporal aspects of GJP coalescence, providing a means of regulating the “when” and “how much” of connexon delivery.

### 3.3. Cx36 Transport Correlates to the Modulation of Cytoskeletal Dynamics 

Pharmacological manipulation of the microtubule network by bath application of 100 µM Colchicine, a microtubule destabilizer, or 20 µM Paclitaxel, a microtubule stabilizer, was used to investigate mechanistic details of Cx36 transport to the GJP. Disruption of actin filaments with Cytochalasin D (3 µM) was included in this investigation since both tubulin and actin-dependent transport has been found at chemical synapses [36] and for the connexin protein Cx43 [37]. Immunoblot analysis and localization studies 48 h post-transfection confirmed Cx36 expression and recruitment into the GJPs at the juxtamembrane (Figure 5A,B). Treatments had no impact on the total Cx36 protein expression (Figure 5A); however, GJP areas were reduced relative to the untreated wild-type (Colchicine: 1.4 ± 0.2 µm^2^, *n* = 15, *p* = 3.00 × 10^−3^; Paclitaxel: 1.4 ± 0.2 µm^2^, *n* = 17, *p* = 7.00 × 10^−3^; Cytochalasin D: 1.7 ± 0.2 µm^2^, *n* = 26, *p* = 0.01) (Figure 5C). GJP formation was uniformly reduced in frequency across all treatments in comparison to the untreated wild-type (Colchicine: 6.7 ± 1.4%, *p* < 0.01 × 10^−1^, Paclitaxel: 10.5 ± 2.2%, *p* < 0.01 × 10^−1^, Cytochalasin D: 7.3 ± 1.3%, *p* < 0.01 × 10^−1^) (Figure 5D). This outcome suggested that an intact and dynamic tubulin cytoskeleton could facilitate connexon delivery to the cell membrane. Additionally, the reduction in GJP formation by Cytochalasin D treatment suggested that the actin cytoskeleton contributed to Cx36 delivery, but mechanistic details were not further investigated.

Co-expression of wild-type Cx36-EGFP and tubulin-mCherry revealed that Cx36 co-localization (overlap coefficient) to the tubulin cytoskeleton was significantly impaired with both Colchicine (WT: 0.59 ± 0.03, *n* = 40; Colchicine: 0.12 ± 0.07, *n* = 21, *p* < 0.01 × 10^−1^) and Paclitaxel treatment (0.44 ± 0.03, *n* = 45, *p* = 3.00 × 10^−3^). Considering that Paclitaxel stabilizes microtubules yet less Cx36 interacted with the tubulin cytoskeleton, we propose that binding of paclitaxel creates a steric hinderance for Cx36 binding. In this way, paclitaxel treatment would act as a competitive inhibitor, controlling the quantity of Cx36 able to bind to tubulin. Our results reaffirmed that Cx36 preferentially interacts with an intact and dynamic tubulin cytoskeleton, promoting transport to the plasma membrane and subsequent coalescence at the GJP. Treatment with Cytochalasin D had no effect on Cx36 co-localization to the tubulin cytoskeleton (0.58 ± 0.01, *n* = 38, *p* = 2.43 × 10^−1^) (Figure 5E, Appendix A). This confirmed that the interaction between Cx36 and the tubulin cytoskeleton was not interdependent on the interactions or stability of the actin cytoskeleton. 

TIRF-trafficking measurements revealed no significant differences in displacement length (WT: 8.7 ± 0.5 µm *n* = 475; Colchicine: 6.5 ± 0.4 µm, *n* = 337, *p* = 3.01 × 10^−1^) or mean speed (WT: 0.1 ± 0.01 µm/s; Colchicine: 0.1 ± 0.01 µm/s, *p* < 0.75 × 10^−1^) with Colchicine treatment in comparison to the untreated wild-type. To consider the effectiveness of drug treatment in an overexpression model and the sensitivity of TIRF detection under these circumstances, the experiments were repeated in single transfected cells for Cx36-EGFP. Here, colchicine application significantly impaired the displacement length (WT: 9.9 ± 0.6 µm, *n* = 271; Colchicine: 7.0 ± 0.6 µm, *n* = 130, *p* = 0.16 × 10^−1^) and mean speed (WT: 0.1 ± 0.01 µm; Colchicine: 0.1 ± 0.01 µm, *p* = 0.02 × 10^−1^) (Appendix A). In double transfected cells, Paclitaxel treatment also impaired vesicular transport (displacement length: 4.5 ± 0.4 µm, *p* < 0.01 × 10^−1^; mean speed: 0.1 ± 0.01 µm/s, *p* < 0.01 × 10^−1^, *n* = 215) emphasizing the importance of a dynamic tubulin cytoskeleton on trafficking vehicles in vivo. Since paclitaxel treatment was sufficient to establish a response in the TIRF field, a single transfected analysis was not further explored. To account for the actin contribution, Cytochalasin D treatment was also included here. Inhibition of vesicular transport was also apparent with Cytochalasin D treatment (displacement, 3.9 ± 0.3 µm, *n* = 237, *p* < 0.01 × 10^−1^; mean speed, 0.1 ± 0.01 µm/s, *p* < 0.01 × 10^−1^), suggesting that actin can serve as a compensatory modality for Cx36 transport and confirming that Cx36 delivery to the PM is both actin and tubulin dependent (Figure 6). 

At the GJP, cytoskeletal interference resulted in a T_1/2_ that was uniformly faster across all treatment groups (WT: 11.5 s, *n* = 28, Colchicine: 1.3 s, *n* = 26; Paclitaxel: 0.6 s, *n* = 24; Cytochalasin D: 0.8 s, *n* = 22). Here, the incorporation of Cx36 into the GJP after actin- or tubulin-cytoskeletal disruption was consistent with diffusion dynamics (Figure 7). Similar to results shown earlier (Figure 4), disruption of the tubulin cytoskeleton with Colchicine increased M*_f_* at the GJP (WT: 25.6 ± 0.6%, Colchicine: 46.4 ± 0.4%, *p* < 0.01 × 10^−2^). No significant difference in recovery was found with Paclitaxel treatment (34.7 ± 0.3%, *p* > 0.99), likely attributable to the direct stabilization of the GJP via tubulin stabilization. Although the GJP remained stable under paclitaxel treatment, it is important to highlight that transport into the GJP was impaired as shown previously (Figure 5 and Figure 6). In this regard, tubulin acts separately, first as a conduit for Cx36 transport and secondly as a GJP stabilizer in the Neuro-2a cell model. Regardless, Cx36 mobility was not completely abolished following tubulin disruption, inferring that the actin cytoskeleton may provide a secondary transport modality. In support, Cytochalasin D treatment significantly increased GJP mobility (55.4 ± 0.4%, *p* < 0.01 × 10^−2^) (Figure 7), confirming that the actin cytoskeleton contributes to Cx36 delivery and stabilization at the GJP.

### 3.4. Characterization of the Tubulin-Binding Motif

A previous report demonstrated that a 26 amino-acid peptide of Cx43 mediates binding to α- and β-tubulin isoforms. This region adopts a helical conformation upon binding to tubulin and is regulated by phosphorylation [38]. Here, we explored whether Cx36 might have a similar binding mechanism using structural modeling of the Cx36 carboxyl-terminal domain. The calmodulin-binding region overlaps the Cx36 CT tubulin-binding region and is known from NMR structural studies to form an amphipathic helix [22]. With this knowledge, a sequence comparison was made between the Cx36 tubulin-binding region and an engineered helix derived from the stathmin and stathmin-like domain (SLD) family of proteins that were known to prevent microtubule formation by binding multiple α- and β-tubulins [39]. From this comparison, we concluded that the Cx36 tubulin-binding region had significant sequence similarities to a portion of the SLD that interacts exclusively with α-tubulin. A sequence comparison, a helical wheel representation of SLD and Cx36, and a molecular view of the SLD-tubulin interaction are presented in Figure 8. For both the engineered SLD and Cx36, the contacts made by their respective amphipathic helices appear to favor four hydrophobic amino acids with possibly a bulkier amino acid at the first position (L47 in SLD and W277 in Cx36), followed by two basic amino acids that make critical ionic contacts near the tubulin α/β interface. An alternative model for a potential Cx36–tubulin interaction was considered using the structure of a centrosomal-P4.1-associated-protein (CPAP) peptide bound to an alpha-beta tubulin heterodimer [40] but was subsequently discounted due to poor sequence and secondary structure similarity.

Sequential alanine-scanning mutagenesis from K279 to G286 was performed to investigate positional roles of individual amino acids in supporting the binding and physiological mechanisms between Cx36 and tubulin. In contrast to the similar expression observed between Cx36Δ279–292-EGFP and WT Cx36 (Figure 2A), all point mutants were expressed at lower and variable levels in comparison to the wild-type protein (Figure 9A). Nonetheless, all mutants formed GJPs at the juxtamembrane (Figure 9B), with a significant reduction in GJP size (K279A: 0.9 ± 0.2 µm^2^, *n* = 7, *p* < 0.01 × 10^−1^; I280A: 1.7 ± 0.3 µm^2^, *n* = 19, *p* = 1.60 × 10^−2^; K281A: 1.1 ± 0.1 µm^2^, *n* = 9, *p* = 1.00 × 10^−3^; L282A: 1.3 ± 0.1 µm^2^, *n* = 16, *p* = 3.00 × 10^−3^; V284A: 1.1 ± 0.2 µm^2^, *n* = 26, *p* < 0.01 × 10^−1^; R285A: 1.6 ± 0.3 µm^2^, *n* = 12, *p* = 2.90 × 10^−2^; G286A: 1.1 ± 0.2 µm^2^, *n* = 10, *p* = 1.00 × 10^−3^) (Figure 9C). Specific targeting of the conserved residues K279A, V284A, and G286A resulted in fewer detected GJPs across cell pairs (*n* = 30, K279A: 8.7 ± 1.6%, *p* < 0.01 × 10^−1^; V284A: 18.4 ± 3.2%, *p* = 8.00 × 10^−3^; G286A: 12.5 ± 2.7%, *p* = 0.01 × 10^−1^) (Figure 9D) as did most non-conserved residues (I280A: 12.1 ± 2.4%, *p* < 0.01 × 10^−1^, K281A: 12.3 ± 2.1%, *p* < 0.01 × 10^−1^, R285A: 16.3 ± 2.8%, *p* = 0.06 × 10^−1^). Cx36-L282A was indistinguishable from the wild-type control (L282A: 24.8 ± 4.9%, *p* = 0.32 × 10^−1^).

Cells co-expressing mCherry-tubulin with Cx36 I280A had a severe reduction in colocalization (overlap coefficient, 0.26 ± 0.05, *n* = 37, *p* < 0.01 × 10^−1^). Amino acids K279A and K281A also had a marked reduction in co-localization to the tubulin cytoskeleton (overlap coefficient, K279A 0.37 ± 0.04, *n* = 36, *p* = 2.10 × 10^−2^; K281A: 0.41 ± 0.05, *n* = 37, *p* = 0.02), suggesting that lysine residues have a significant role in direct binding or anchoring to tubulin. The remaining mutants were indistinguishable from the wild-type (overlap coefficient, L282A: 0.41 ± 0.06, *n* = 36, *p* = 5.30 × 10^−2^; V284A: 0.48 ± 0.04, *n* = 43, *p* = 3.18 × 10^−1^; R285A: 0.49 ± 0.04, *n* = 26, *p* = 3.62 × 10^−1^; G285A: 0.44 ± 0.05, *n* = 26, *p* = 2.19 × 10^−1^) (Figure 9E and Appendix A). Based on the 3D structural model and co-localization results, we propose that Cx36 adopts a similar binding mechanism to SLD; however, in vivo, the downstream adjacent amino acids K279A, I280, and K281 appear to support the majority of interactions. 

To investigate the impact of positional amino-acid alterations on vesicular transport, displacement length and speed were determined by TIRF microscopy. A reduction in displacement length was observed across select mutants tested (K281A: 5.8 ± 0.5 µm, *n* = 273, *p* < 0.01 × 10^−1^; R285A: 9.8 ± 0.6 µm, *n* = 302, *p* < 0.01 × 10^−1^; G286A: 9.6 ± 0.6 µm, *n* = 286, *p* = 0.02 × 10^−1^). All else were indistinguishable from the WT (K279A: 8.9 ± 0.6 µm, *n* = 390, *p* = 5.94^−1^; I280A: 7.8 ± 0.6 µm, *n* = 237, *p* = 8.80^−1^; L282A: 7.6 ± 0.5 µm, *n* = 244, *p* = 9.57 × 10^−1^; V284A: 8.7 ± 0.5 µm, *n* = 423, *p* = 1.11 × 10^−1^) (Figure 10A). Similarly, mean speed was reduced exclusively for K281A, R285A, and G286A mutations (K281A: 0.1 ± 0.01 µm/s, *p* < 0.01 × 10^−1^; R285A: 0.1 ± 0.01 µm/s, *p* < 0.01 × 10^−1^; G286A: 0.1 ± 0.01 µm/s, *p* = 1.70 × 10^−2^), and all else were indistinguishable from the WT (K279A: 0.1 ± 0.01 µm/s, *p* = 4.46 × 10^−1^; I280A: 0.1 ± 0.01 µm/s, *p* = 6.01 × 10^−1^; L282A: 0.1 ± 0.01 µm/s, *p* = 6.40 × 10^−1^; V284A: 0.1 ± 0.01 µm/s*, p* = 2.38 × 10^−1^) (Figure 10B). Together with GJP size, formation, and co-localization ability, our results have identified segregated functionality within the binding motif, specific to either the binding or trafficking mechanisms.

Similar to the deletion mutant Cx36Δ279–292, a faster T_1/2_ was recorded for several mutants in comparison to the wild-type control in FRAP investigations (WT: 11.5s, K279A: 2.2 s, *n* = 10; I280A: 0.9 s, *n* = 20; L282A: 3.5s, *n* = 23; R285A: 47.8 s, *n* = 22). Furthermore, these select mutants displayed a significantly higher M*_f_* (WT: 25.6 ± 0.6%, K279A: 41.5 ± 0.5%, *p* < 0.01 × 10^−2^; I280A: 46.2 ± 0.5%, *p* < 0.01 × 10^−2^; L282A: 40.6 ± 0.6%, *p* = 0.07 × 10^−2^; R285A: 44.3 ± 13.8, *p* < 0.01 × 10^−2^). Our results confirmed that single amino-acid changes affected the mobility of Cx36, again, highlighting the sensitivity of this motif. Amongst these mutants, a less controlled mechanism of GJP incorporation, through the loss of protein-binding partnerships, was being promoted. Consistent with our previous model (Figure 8) outlining K281 as critical in the tubulin–Cx36-binding mechanism, this positional mutation resulted in the highest increase in mobility (60.6 ± 10.4%, *n* = 15, *p* < 0.01 × 10^−2^), paired with an excessive increase in T_1/2_ (32.18 s) (Figure 11). Please note that a curve fit for V284A and G286A mutations using a mono-exponential association equation consistently for all other mutants was not possible and a detailed analysis of M*_f_* and T_1/2_ could not be provided. However, recovery ability was significantly different from that of the wild-type control (V284A: *n* = 19, *p* < 0.01 × 10^−2^; G286A: *n* = 14, *p* < 0.01 × 10^−2^) (Figure 11).

## 4. Discussion

This research uses a neuroblastoma model of synaptic plasticity to explore the interactions between Cx36 and tubulin. By sequence similarity to Cx43 [20], a tubulin-binding motif was identified in the carboxy-terminal tail of Cx36. In experimental support, the previously reported binding of tubulin with the carboxyl-terminal tail of Cx36 [19] was confirmed independently here using a BioID assay. The Cx36-binding motif is highly conserved and unique among connexin isoforms as it shares its interaction surface amongst tubulin-, CaM-, and CaMKII-binding partners. Typically, CaM-binding motifs in other connexins are located in different positions in the amino-terminus (Cx32), cytoplasmic loop (Cx43, Cx44, and Cx50), or nonoverlapping region within the CT domain (Cx32) [12,41,42,43,44,45]. Binding of CaMKII to connexins other than Cx36 has not been reported.

The Cx36–tubulin-binding motif differs from other microtubule-binding motifs found in proline-rich regions of several microtubule-associated proteins (MAPs) or the motor proteins kinesin and dynein. For example, the microtubule-binding motif in both motor proteins has a P-X_6_-E-X_4_-L core consensus sequence, surrounded by several conserved polar, hydrophobic, and charged amino acids [46,47]. Similarly, the tubulin-binding motif of Cx36 is enriched in hydrophobic (isoleucine, leucine, arginine, and valine) and electrically charged basic (arginine and lysine) amino acids. However, the absence of a proline-rich region appears to be a significant and unique property of the Cx36–tubulin motif.

Three-dimensional structural modeling of the Cx36–tubulin-binding site to the engineered stathmin-like domain (SLD) predicted similarities between Cx36 and SLD. The Cx36 carboxyl terminus likely adopts a helical structure upon binding with tubulin similar to previous reports on Cx43–tubulin interactions [38]. The previously resolved NMR structure of the Cx36–CaM-binding complex suggests that a helical structure is formed as early as interaction with calcium-activated CaM occurs in the ER–Golgi complex. Siu et al. have also shown that CaM binding strongly relies on access to W277 and intracellular calcium [22]. The new experimental evidence supports roles of individual amino acids downstream of W277, with I280 and the adjacent amino acids K279 and K281 critical in the Cx36–tubulin-binding mechanism as demonstrated by co-localization studies and consistent with the structural model. Noteworthily, phosphorylation of S293 [48,49,50] downstream of the overlapping tubulin/CaM/CaMKII-binding site decreases GJ communication, suggesting that competitive and cooperative mechanisms exist side-by-side in a protein domain encompassing less than 20 amino acids [51]. Consistent with other studies [52,53,54,55,56,57], we anticipate that salt bridges formed by the lysine residues contribute to protein stability and specificity, driving molecular recognition and catalysis between protein interfaces. We also found that site-directed alanine mutagenesis produced variability in protein expression levels that were not previously observed with the expression of Cx36Δ279–292. Since large amino-acid deletions in the loop, helix, and terminal positions are generally more tolerated than substitutions [58,59], we concluded that the Cx36 tertiary structure and functionality were likely compromised, leading to expression variability. Again, we believe this advocated for the significance of positional non-covalent interactions to sustain both the protein itself and protein–protein interfaces.

The Cx36Δ279–292 mutant exhibited less co-localization to the ER–Golgi complex but more co-localization with caveolin-1 vesicles. Although the ER stress response was not explored in this study, equal protein expression of wild-type and mutant proteins as well as lack of signs of protein degradation suggest that involvement of the endoplasmic reticulum-associated degradation (ERAD) pathway is minor. We propose a mechanism in which the mutant protein is prematurely released from the ER–Golgi complex prior to its packaging into transport vesicles. Since we have demonstrated that Cx36Δ279–292 is still transport competent, a different mechanism of transport may be favored. Interaction with actin is a possibility. The increase in the association between Cx36Δ279–292 and caveolin-1-positive vesicles could be an indication of alternative transport pathways.

The pharmacological manipulation of the microtubule network and genetic manipulation of the tubulin-binding motif confirmed that the trafficking of Cx36-carrying vesicles and the formation of GJPs were critically dependent on an intact and functionally dynamic cytoskeleton. The processes observed appear to follow the generally accepted chain of events in which connexons packaged in vesicles emerge from the Golgi reach the cell membrane via microtubules at multiple nonrandom insertion sites [28,33,34]. After that, hemichannels form clusters of gap junctions known as plaques at cell–cell borders. In addition to the modest similarities in the binding motif, the aforementioned transport mechanism parallels the reports for Cx43 [28,33].

Lauf et al. previously described the directed transport of Cx43 to the plasma membrane, demonstrating mobility of approximately 70%, before incorporation into GJPs. At GJPs, Cx43 has been described as exhibiting either low or high mobility states at the lateral ends. Categorization of the mobility state was found to be independent of GJP size but, rather, to be determined by the C-terminal domain influencing channel density, protein interaction candidates, or both [60]. Our studies have confirmed unique differences between Cx43 and Cx36 kinetics both at the plasma membrane and GJP. In this research, we report mobility of wild-type Cx36 as approximately 70% at the plasma membrane, comparable to Lauf et al., and as 25% at the GJP. We suspect that mobility in the PM contributes to the fine-tuning of Cx36 incorporation, especially under activity-dependent circumstances. In a related study [50], FRAP microscopy of GJPs formed by Cx36-HaloTag fusion proteins was studied in HeLa cells. Here, the initial M_f_ was reported to be 56% and subsequently decreased to 41% after sequential bleaching. Wang et al. also found half-time of recovery to be 1.55 ± 0.22 s [50]. In this study, the carboxyl terminus of Cx36 was fused with the GFP protein tag, increasing the molecular mass by 27 kDa. We anticipate that the lower mobile fraction and longer half-time of recovery differences observed between our wild-type protein and the Cx36-HaloTag can be attributed to the large C-terminal GFP tag, decreasing the mobility potential of Cx36. Nevertheless, we consistently reported considerably shorter half-times of recoveries upon manipulation of the tubulin cytoskeleton or interaction thereof. Since a faster T_1/2_ is typically reflective of weaker binding mechanisms, incorporation of Cx36 at the GJP under these circumstances was likely mediated by diffusion rather than a controlled transport mechanism.

Theoretical and practical considerations suggest that other transport mechanisms must exist alongside tubulin-dependent transport. In neurons, microtubules participate in axonal vesicular transport and tubulin entry into dendritic spines is activity-dependent or associated with development [61]. Lateral diffusion in the membrane as mechanisms for transport of Cx36 into dendritic spines cannot explain temporal aspects of the plasticity of electrical synapses. Here, experimental evidence showed that treatment conditions targeting tubulin or the tubulin-binding motif reduced the GJP area and formation but that the formation of GJPs at the juxtamembrane was not wholly abolished. We propose a second, actin-dependent transport and GJP formation process as suggested by Lynn et al. [62], similar to actin-dependent transport of receptor and channel proteins into synaptic spines [63,64,65,66]. Binding of Cx36 to actin directly or indirectly through adaptor proteins has not been addressed in this study. Although previous reports indicated that interaction with the actin cytoskeleton was involved in the turnover and invagination endocytosis of gap junctions [67], a more recent study on Cx30 proposed that actin is involved in the anchoring of connexons and short distance transport rather than in the facilitation of GJ internalization [27]. Follow-up studies resolving details of this second transport mechanism will help to understand how molecular and functional asymmetry is created at a vertebrate electrical synapse. 

Wayakanon et al. investigated the consequences of GJ transport and internalization of various CT truncated Cx43 mutants. Deletion of amino acids 235–242, which corresponds to the tubulin-binding motif, resulted in the absence of GJPs [68]. In contrast, our results demonstrated the retention of GJP formation and communication followed deletion of the tubulin-binding motif in Cx36, albeit at reduced efficiencies. Similarly, deletion of tubulin-binding motif in the N2RB subunit of altered NMDA receptors reduced but not abolished transport to the synapse [69]. Further, cell type-based differences will affect transport and internalization of both electrical and chemical modalities. We propose that neurons may have developed specialized transport mechanisms supporting on-demand plasticity. Our results point to actin as a secondary conduit for Cx36 transport, similar to NMDA interactions with actin filaments [70]. Further investigations would be required to determine whether actin- and tubulin-binding efficiencies are the basis for a tug of war between two competing or cooperating pathways.

What is the physiological relevance of Cx36 interaction with microtubules? Cx36 has one of the lowest voltage sensitivities (half-inactivation voltage ±75 mV) and single-channel conductance (10–15 pS) among other connexin isoforms [71]. The detection of Cx36 GJP in the nervous system had been elusive for a long time due to the small size [72,73]. Arguably, the small GJP size and Cx36-specific gating properties suggest that a precise electrical coupling can be achieved through the tight regulation of the number of channels present in the gap-junction plaque [71]. This mechanism was attributed to tubulin-dependent trafficking and was complemented by the phosphorylation via CaM/CaMKII [4] and protein kinase A (PKA) [50,51,74,75,76]. We acknowledge that such a refined transport mechanism could serve both to enhance neuronal plasticity and/or to reduce excitotoxicity and subsequent neuronal cell death, both of which are influenced by Cx36 GJ coupling. Our results point to the significance of the Cx36–tubulin interaction; tubulin is a fundamental component in the modulation of neuronal synapses via channel abundance and stability.

Our investigation provides insight on a critical step towards molecular and functional asymmetry at a vertebrate electrical synapse. Tubulin-dependent transport connects two major bookends of the life cycle of Cx36. Before interaction with tubulin, direct interaction with CaM occurs primarily at the ER/Golgi complex [22]. Vesicular transport to the GJP involves tubulin and most likely another transport system accounting for the structural and functional specialization of pre- and postsynaptic compartments of soma-somatic, dendro-dendritic, dendro-somatic, and axon-dendritic contact sites in mature neurons where Cx36 is expressed [23,24]. Outcomes of this research extend to previously recognized similarities of both NMDA receptors and Cx36 interaction with CaM/CaMKII. Similar to Cx36, NMDA receptor channel subunit CTs bind tubulin dimers or soluble forms of tubulin. The efficient modulation of microtubule dynamics by the NR1 and NR2 cytoplasmic domains suggests an interaction of the receptor and the subsynaptic cytoskeletal network that may play a role during morphological and functional adaptations, as observed during synaptogenesis and in adult CNS plasticity. Importantly, results have shown that tubulin-dependent trafficking of Cx36 is part of the molecular machinery potentiating electrical synaptic strength. In this regard, the activity-dependent modulation of the cytoskeleton controls the formation and plasticity of electrical synapses.

## Figures and Tables

**Figure 1 cells-08-01146-f001:**
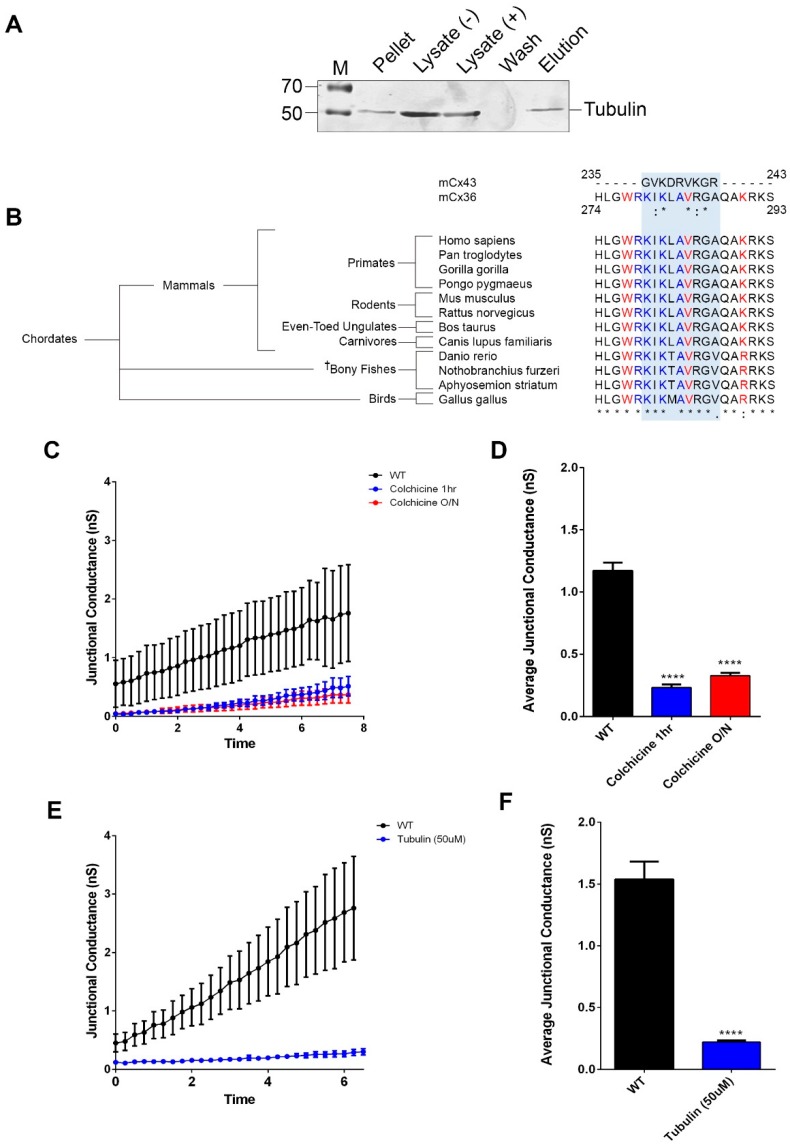
Tubulin interaction at the Cx36 carboxyl terminus potentiates electrical plasticity. (**A**) Full-length Cx36 protein was expressed as a fusion protein with the BirA* ligase to perform a BioID assay using Neuro-2a cell lysate. Lysates were subjected to western blot analysis before (−) and after (+) the addition of streptavidin-conjugated beads. Immunodetection identified tubulin as an interaction candidate of Cx36. (**B**) Sequence alignment of the tubulin-binding motif confirmed in Cx43 elucidated the potential Cx36 interface, localized to the Cx36 carboxyl terminus (CT). The putative binding motif of Cx36, shaded in blue, demonstrated robust conservation across orthologs belonging to the major vertebrate Chordata subphylum. The † symbol indicates the selective use of the Cx35b protein corresponding to the *gjd2b* gene. Conserved residues within the Cx36 CT are indicated by the asterisks. Both CaM- and CaMKII-binding sites appeared to share some overlap with the tubulin-binding motif, potentially indicating competitive or cohesive binding mechanisms. The CaMKII-binding motif is outlined in blue, and the CaM motif is outlined in red. (**C**,**D**) Neuro-2a cells cultured in a bathing solution supplemented with 100 µM Colchicine for 1 h (*n* = 5) or 24 h (*n* = 6) demonstrated a significant reduction in gap-junctional (GJ) conductance relative to the untreated wild-type (*n* = 6). Data are the mean ± SEM; **** *p* < 0.0001, Kruskal–Wallis test; each sample was compared to the untreated wild-type (WT). (**E**,**F**) Intracellular perfusion with monomeric tubulin (50 µM; *n* = 5) through the recording electrode was equally sufficient to block an increase in junctional currents in comparison to the untreated WT (*n* = 4). Data are the mean ± SEM; **** *p* < 0.0001, Mann–Whitney test.

**Figure 2 cells-08-01146-f002:**
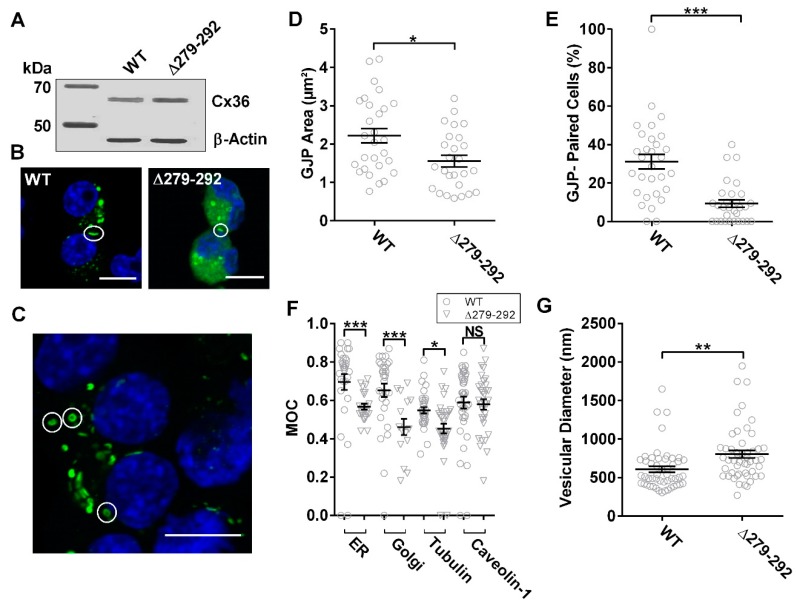
Loss of interaction between tubulin and Cx36 promotes intracellular localization. (**A**) Immunoblot analysis of Neuro-2a cells transiently transfected to express EGFP-tagged wild-type and ∆279–292 Cx36 constructs. Fusion proteins were detected using an anti-EGFP antibody confirming expression. An anti-β-actin antibody served as the loading control; protein standard is denoted in kDa. (**B**,**C**) Neuro-2a cells were fixed with formaldehyde and mounted onto slides with DAPI containing media 48 h post-transfection. Cx36 proteins are shown in green. In Figure 2B, examples of Cx36 gap-junction plaques are indicated by circles, and in Figure 2C, examples of annular junctions formed in Cx36∆279–292-expressing cells are indicated by circles. Scale bar: 10 µm, nuclear DAPI staining in blue. Quantification of gap-junction plaque (GJP) size (**D**) and incidences of GJP-pairings (**E**) demonstrated a significant reduction in connexon recruitment in cells expressing the tubulin-binding deficient mutant Cx36∆279–292, indicative of trafficking impairments. Data are mean ± SEM; for GJP area, sample sizes (n) were 28 and 26 for Cx36 WT and ∆279–292 proteins, respectively. For GJP-paired cells, *n* = 30, where 10 images were collected in 3 independent experiments. * *p* < 0.05, *** *p* < 0.0001, Kruskal–Wallis test. (**F**) Co-localization of Cx36 vesicles to organelles was quantified by Mander’s overlap coefficient to decipher its intracellular localization. Relative to the wild-type, expression of the tubulin-binding deficient protein resulted in a reduction in ER, Golgi, and tubulin co-localization; caveolin co-localization remained unaffected. Data are mean ± SEM; sample sizes (*n*) of WT Cx36 were *n* = 30 for the ER, *n* = 30 for the Golgi, *n* = 34 for tubulin, and *n* = 40 for Caveolin-1. Sample sizes of Cx36∆279–292 were *n* = 35, *n* = 15, *n* = 36, and *n* = 33 for colocalization of the ER, Golgi, tubulin, and Caveolin-1 proteins, respectively. * *p* < 0.05, *** *p* < 0.0001, Kruskal–Wallis test; each sample was compared to its corresponding wild-type values. (**G**) Cx36 vesicular diameter of cells co-transfected with caveolin-1 demonstrated an increase in size with mutant expression over the wild-type. Data are mean ± SEM; sample size (*n*) of WT Cx36 was *n* = 50 vesicles. ** *p* < 0.001, Kruskal–Wallis test; data were collected from 3 independent experiments.

**Figure 3 cells-08-01146-f003:**
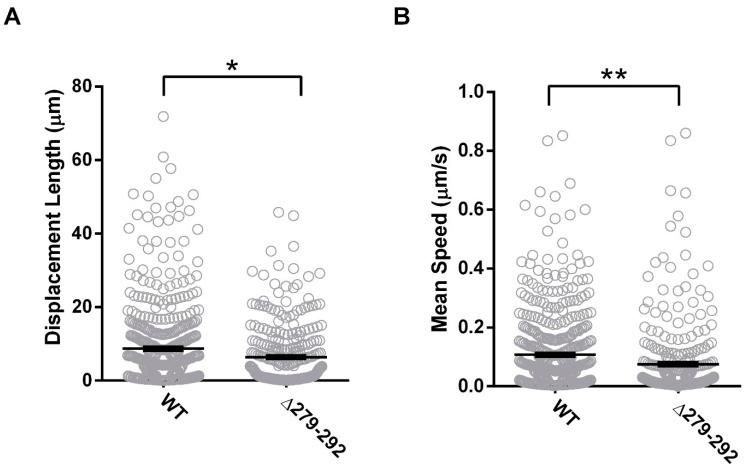
Tubulin interaction mediates the spatiotemporal delivery of Cx36 vesicles within the submembrane. In Cx36-EGFP and tubulin-mCherry co-transfected cells, submembrane vesicular transport was monitored using Total Internal Reflection Fluorescence (TIRF) microscopy. Mutant expression reduced trafficking as determined by the reduction in (**A**) displacement (µm) and (**B**) mean speed (µm/s). Data are mean ± SEM; sample sizes (*n*) of WT Cx36 were *n* = 475 and *n* = 304 for Cx36 ∆279–292 vesicles. * *p* < 0.05, ** *p* < 0.001, Kruskal–Wallis test.

**Figure 4 cells-08-01146-f004:**
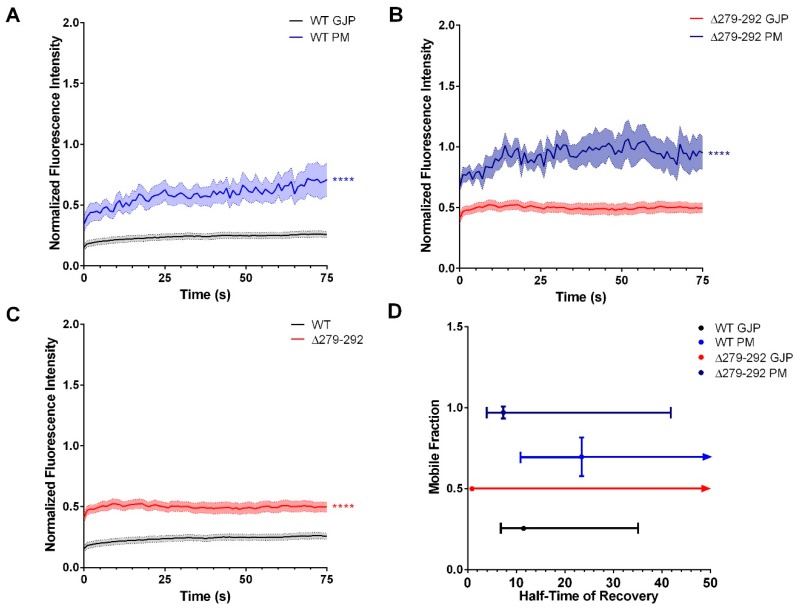
Cx36 connexons are initially delivered to the plasma membrane via the tubulin cytoskeleton and diffuse laterally to the GJP. Fluorescence Recovery After Photobleaching (FRAP) microscopy was used to bleach the non-junctional plasma membrane (PM) and gap-junction plaque (GJP) regions of interest (ROI). The average fluorescence intensity of each ROI during the recovery phase was normalized to the pre-bleached intensity values and plotted to quantify the relative recovery of fluorescence. In cells expressing Cx36 wild-type (**A**) and ∆279–292 (**B**), the plasma membrane (PM) recovered more than at the GJP. Recovery curves are the mean ± SEM. Sample sizes (n) of WT Cx36 were n = 28 for the GJP and *n* = 12 for the PM. For Cx36∆279–292, the sample sizes were *n* = 22 for the GJP and *n* = 10 for the PM. **** *p* < 0.0001, Kruskal–Wallis test. (**C**) An overlap of the GJP curves for the WT (in Figure 4A) and ∆279–292 (in Figure 4B). (**D**) Plot of the mobile fraction (M*_f_*) against the half-time of recovery (T_1/2_) extrapolated from the Cx36 WT (in Figure 4A) and ∆279–292 (in Figure B) recovery curves. The mobility, as indicated by M*_f_*, of both WT and Δ279–292 GJPs were lower than its respective PM ROI, suggesting that the GJP was more stable. Mutant GJPs were significantly more mobile than the WT control. Additionally, T_1/2_ for ∆279–292 was faster at the GJP and PM in comparison to the wild-type, indicating diffusion was favored over a controlled trafficking mechanism. Data are the mean ± CI.

**Figure 5 cells-08-01146-f005:**
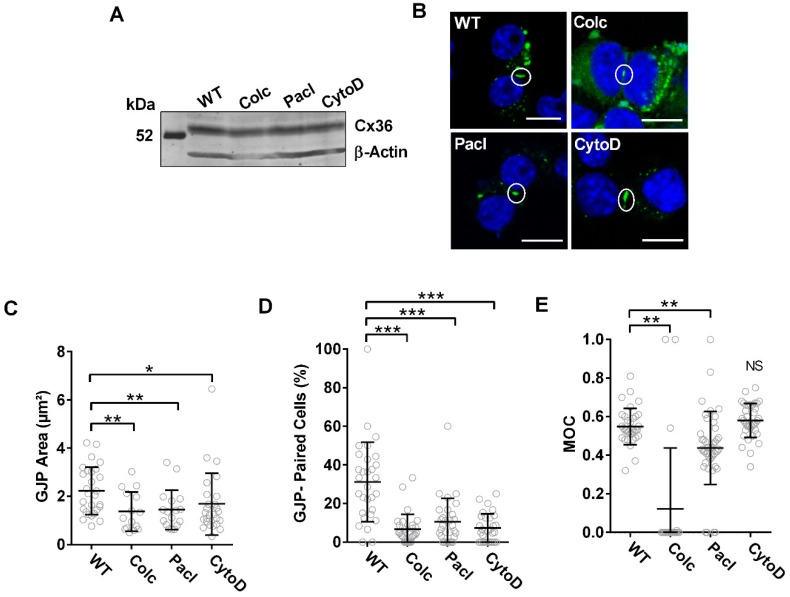
Pharmacological manipulation of the cytoskeletal network inhibits Cx36 recruitment. (**A**) Immunoblot analysis of Neuro-2a cells were transiently transfected to express wild-type Cx36-EGFP and subsequently treated with Colchicine (Colc), Paclitaxel (Pacl), or Cytochalasin D (CytoD). No differences in the expression levels were noted. Cx36 protein was detected using an anti-EGFP antibody, and an anti-β-actin antibody was used to detect actin as the loading control. Protein standard is denoted in kDa. (**B**) Neuro-2a cells were fixed with formaldehyde and mounted onto slides with DAPI containing media 48 h post-transfection. Cx36 proteins are shown in green. Formation of GJPs in vivo under pharmacological manipulation of the tubulin cytoskeleton (Colchicine and Paclitaxel) or the actin cytoskeleton (Cytochalasin D) are shown. Circles indicate GJPs. Scale bar: 10 µm, nuclear DAPI staining in blue. Quantification of GJP area (**C**) and incidences of GJP formation (**D**) revealed severe impairments under all pharmacological treatments tested. Data are mean ± SEM. For GJP area, sample sizes (*n*) were 28 for the WT, *n* = 15 for Colchicine, *n* = 17 for Paclitaxel, and *n* = 26 for Cytochalasin D. For GJP-paired cells, *n* = 30, where 10 images were collected in 3 independent experiments. * *p* < 0.05, ** *p* < 0.001, *** *p* < 0.0001, Kruskal–Wallis test; all samples were compared to the WT control. (**E**) Co-localization studies were performed between Cx36 and the tubulin cytoskeleton; co-localization was expressed using Mander’s overlap coefficient. Treatment with Colchicine and Paclitaxel reduced Cx36-tubulin co-localization, suggesting tubulin must be intact to support sufficient Cx36 interaction. Treatment with Cytochalasin D had no impact on the co-localization of Cx36 to the tubulin. Data are mean ± SEM. Sample size (*n*) indicated the number of cells measured for tubulin co-localization and were *n* = 34 for the WT, *n* = 21 for the Colchicine, *n* = 45 for Paclitaxel, and *n* = 38 for Cytochalasin D. ** *p* < 0.001, Kruskal–Wallis test; all samples were compared to its corresponding WT values.

**Figure 6 cells-08-01146-f006:**
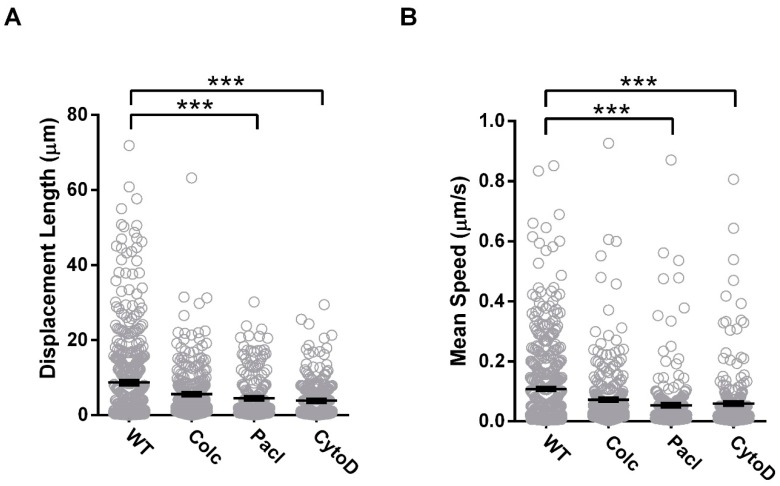
Transport and incorporation of Cx36 into the GJP is dependent on intact actin and tubulin cytoskeletons. Vesicular transport within the submembrane region was monitored using TIRF microscopy to determine the (**A**) displacement (µm) and (**B**) speed (µm/s) of Cx36 vesicles. Colchicine (Colc) did not significantly affect Cx36 vesicular trafficking. However, both Paclitaxel (Pacl) and Cytochalasin D (CytoD) treatment resulted in significant reduction in trafficking when administered to Neuro-2a cells. Data are mean ± SEM; sample sizes (*n*) were *n* = 475 for WT, *n* = 294 for Colchicine, *n* = 215 for Paclitaxel, and *n* = 237 for Cytochalasin D treated vesicles. *** *p* < 0.0001, Kruskal–Wallis test; all samples were compared to the WT control.

**Figure 7 cells-08-01146-f007:**
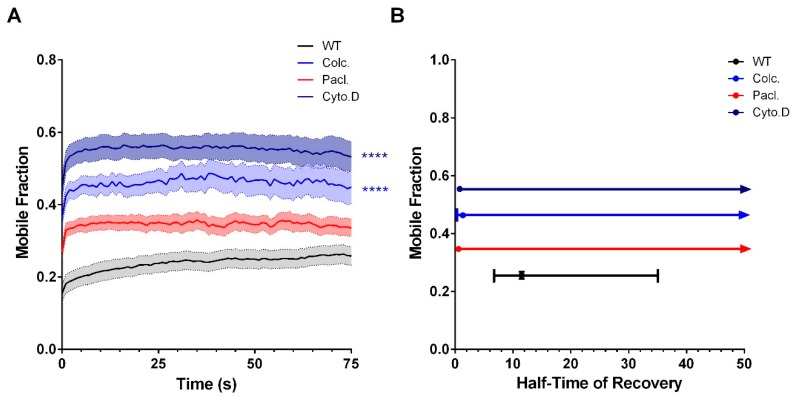
Cytoskeletal manipulation alters Cx36 delivery at the GJP. Recovery of fluorescence curves (**A**) and corresponding plots for M_f_ against T_1/2_ (**B**) demonstrated the relationship between the mobile fraction and half-time of recovery. The M_f_ and T_1/2_ under Colchicine (Colc) and Paclitaxel (Pacl) treatment demonstrated a significant reduction. While the T_1/2_ remained lower than the untreated wild-type, inhibition of the actin cytoskeleton with Cytochalasin D (CytoD) resulted in an increase in the mobile fraction, indicating that GJPs became more fluid and suggesting that actin may play a role in fine-tuning Cx36 incorporation at the GJP. Recovery curves are the mean ± SEM and mean ± CI for the XY plots of mobile fraction against half-time of recovery. Sample sizes (*n*) were n = 28 for the WT, *n* = 26 for Colchicine, *n* = 24 for Paclitaxel, and *n* = 22 for Cytochalasin D treated GJPs. **** *p* < 0.0001, Kruskal–Wallis test; all samples were compared to the wild-type control.

**Figure 8 cells-08-01146-f008:**
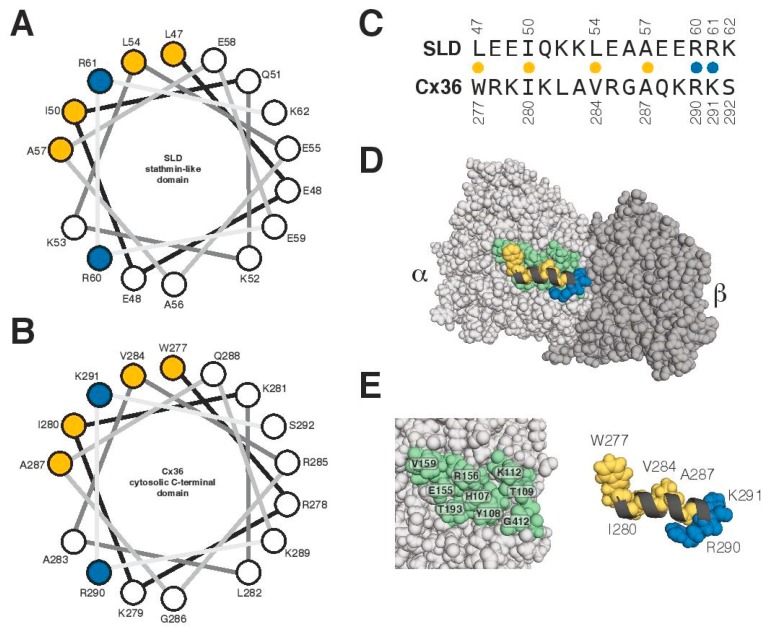
Structural modeling of the CT: A model of the interaction between Cx36 and tubulin. When sequence and secondary structure similarity are considered together, the crystal structure of an engineered stathmin-like domain (SLD) and an alpha/beta tubulin protofilament (PDB: 1Z2B) emerges as a possible template to consider an analogous Cx36 interaction. (**A**,**B**) Helical wheel representations of the relevant regions of the SLD (47–62) and Cx36 (277–292) with key amino acids colored at the interface according to either hydrophobicity (orange) or charge (blue). (**C**) A sequence comparison following the same scheme as the helical wheel. (**D**,**E**) Three views of a Cx36/tubulin model highlighting the potential contacts made between Cx36 (black ribbon) and alpha tubulin (light grey and green). Models were created in PyMOL v1.86.

**Figure 9 cells-08-01146-f009:**
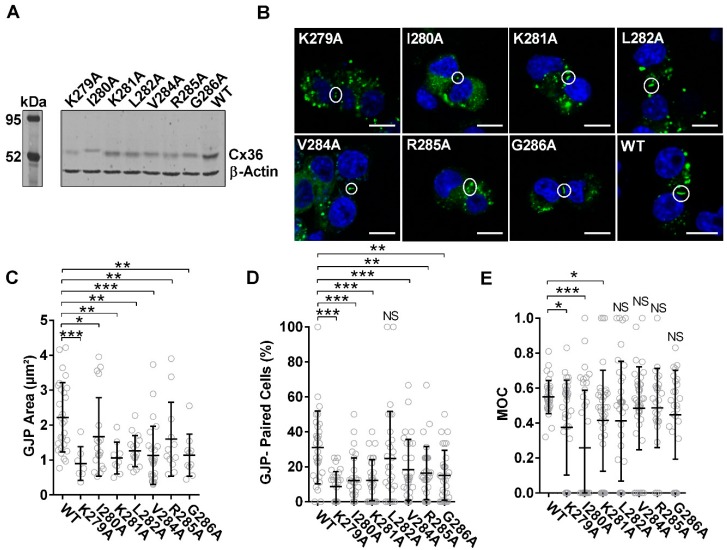
The Cx36-tubulin protein interface is sensitive to targeting single amino acids. (**A**) The tubulin-binding motif (corresponding to amino acids Lys279–G286) was targeted by alanine-scanning mutagenesis. Immunoblot analysis of Neuro-2a cells transiently transfected to express EGFP-tagged wild-type and mutant constructs reveals variable expression. Proteins were detected using an anti-EGFP antibody. An anti-β-actin antibody was used as a loading control; protein standard is denoted in kDa. (**B**) Neuro-2a cells were fixed with formaldehyde and mounted onto slides with DAPI containing media 48 h post-transfection. Cx36 proteins are shown in green. Formation of GJPs at the juxtamembrane was confirmed in vivo using confocal microscopy post-mutagenesis. Boxes indicate GJPs. Scale bar: 10 µm, nuclear DAPI staining in blue. (**C**) Quantitatively, a severe reduction in GJP size was observed across all Cx36 mutants. Data are mean ± SEM. Sample sizes (*n*) were the following, WT: 28, K279A: 7, I280A: 19, K281A: 9, L282A: 16, V284A: 26, R285A: 12, and G296A: 10 GJPs. * *p* < 0.05, ** *p* < 0.001, *** *p* < 0.0001, Kruskal–Wallis test; all samples were compared to the WT control. (**D**) Formation of GJPs between cell–cell contacts was impacted when both the residues conserved between Cx36 and Cx43, Cx36-K279, -V285, and -G286 and non-conserved residues Cx36-I280, -K281, and G285 were targeted. Sample size for GJP-paired cells was *n* = 30, where 10 images were collected in 3 independent experiments. ** *p* < 0.001, *** *p* < 0.0001, Kruskal–Wallis test; all samples were compared to the WT control. (**E**) Co-localization to the tubulin cytoskeleton was reduced with Cx36 K279A, I280A, and K281A expression. Sample sizes (n) were the following: WT: 34, K279A: 36, I280A: 37, K281A: 37, L282A: 36, V284A: 43, R285A: 26, and G296A: 26 cells. * *p* < 0.05, *** *p* < 0.0001, Kruskal–Wallis test; all samples were compared to the WT control.

**Figure 10 cells-08-01146-f010:**
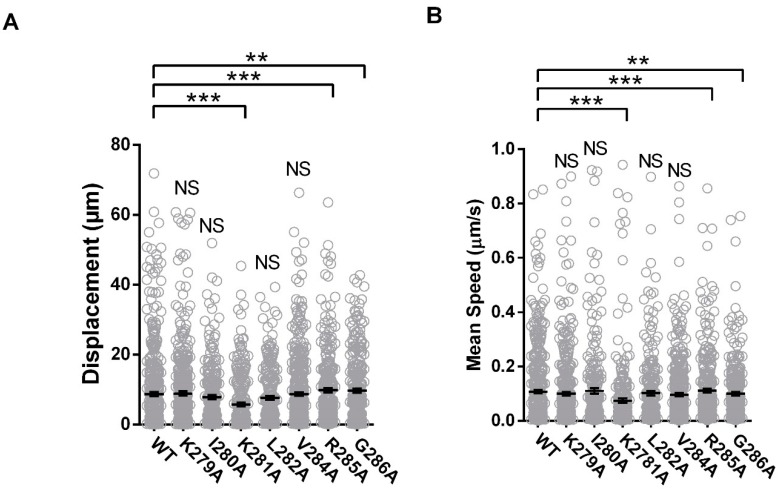
Reduced trafficking of Cx36 vesicles post-alanine mutagenesis. Submembrane vesicular transport of the point mutant constructs was resolved using TIRF microscopy in Neuro-2a cells. Cx36 K281A, R285A, and G286 mutants significantly disrupted displacement (**A**) and mean speed (**B**). As such, positional interference of the tubulin-binding motif was sufficient to disrupt spatiotemporal aspects of protein trafficking. Data are mean ± SEM. Sample sizes (*n*) were the following: WT: 475, K279A: 390, I280A: 237, K281A: 273, L282A: 244, V284A: 423, R285A: 302, and G296A: 286 vesicles. ** *p* < 0.001, *** *p* < 0.0001, Kruskal–Wallis test; all samples were compared to the WT control.

**Figure 11 cells-08-01146-f011:**
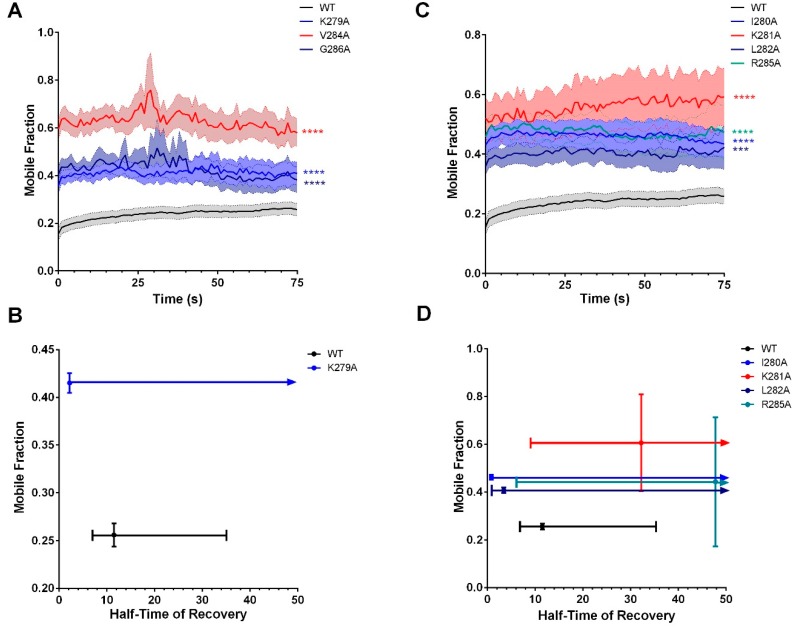
Recovery of Cx36 at the GJP was impaired after targeting individual amino acids within the tubulin-binding motif. Repopulation of connexons at the gap-junction plaque was significantly impaired in all mutants targeted in the site-directed alanine scan of the tubulin-binding motif. (**A**,**B**) Recovery plot and corresponding XY scatterplot illustrating the relationship between the mobile fraction (%) and half-time of recovery (s) of the residues conserved between Cx36 and Cx43 in the tubulin-binding motif. Recovery curves are the mean ± SEM and mean ± CI for the XY plots of mobile fraction against half-time of recovery. Sample sizes (*n*) were *n* = 28 for the WT, *n* = 10 for K279A, *n* = 19 for V284A, and *n* = 14 for G286A GJPs. **** *p* < 0.0001, Kruskal–Wallis test; all samples were compared to the wild-type control. Please note curve fitting for V284A and G286A was not possible; therefore, neither are diagramed here. (**C**,**D**) Recovery plot and corresponding M*_f_* vs. T_1/2_ plot non-conserved residues. Since curve fitting was unsuccessful for Cx36-R285A, it was not reported in the M*_f_* vs. T_1/2_ scatterplot. Deviations in the M*_f_* and T_1/2_ suggest that alternations to the tubulin-binding motif were sufficient to provoke phenotypes similar to the complete ablation of the tubulin-binding motif. Recovery curves are the mean ± SEM and mean ± CI for the XY plots of mobile fraction against half-time of recovery. Sample sizes (*n*) were *n* = 28 for the WT, *n* = 20 for I280A, *n* = 15 K281A, *n* = 23 for L282A, and *n* = 22 for R285A GJPs. *** *p* < 0.001, **** *p* < 0.0001, Kruskal–Wallis test; all samples were compared to the wild-type control.

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
