# Peer review of "Tubulin-Dependent Transport of Connexin-36 Potentiates the Size and Strength of Electrical Synapses"

_cells, 2019, doi:10.3390/cells8101146_

Round 1
Reviewer 1 Report
The authors described tubulin-dependent Cx36 transport implicated in the size and strength of electrical synapses. The authors confirmed the interaction between tubulin and Cx36 which has both tubulin binding motif and CaMKII/CaM binding motif. Owing to loss of tubulin interaction, carboxy-terminal deleted mutant Cx36 impaired electrical plasticity and Cx36 gap junction localization and transport of Neuro-2A cells. By using 3D structural modeling, the authors identified tubulin interface of Cx36 that modulated Cx36 gap junction localization and transport. The approach and the finding are very interesting, but this reviewer has several comments as described below:
Major comments
Figure 2-4. Since the carboxy-terminal (CT) of Cxs might regulate connexon and gap junction formation, the experiment of CT-deleted mutant Cx36 is not enough to investigate impact of tubulin-interaction. The inhibition assay of tubulin-Cx36 interaction by peptide and antibody will support the authors results.
The authors should mention about behavior of CT-deleted mutant Cx43 which has no tubulin binding motif and tubulin motif deleted NMDA receptor or NR2B for comparison with Cx36.
Figure 2F and Figure 3, as no references and no appropriate control, this reviewer cannot fully understand the results that CT-deleted mutant Cx36 alters Cx36 localization by loss of interaction with tubulin.
Figure 5-7, appropriate negative control such as Cx43 is required.
It has been thought that altered Cxs expression and function might involve in tumorigenesis and transformation. It is better to describe not only Neuro-2a cells, but also primary cells or other cell lines.
All experiments were performed by in vitro models. The authors need to discuss whether the results in here has physiologically important.
Minor comments
L58, what is wildtype an mutant Cx36? Figure 1, before (-) and after (+) might lead to a misunderstanding. L421, (e) should be replaced to (E).
Author Response
Reviewer 1
Comments and Suggestions for Authors
The authors described tubulin-dependent Cx36 transport implicated in the size and strength of electrical synapses. The authors confirmed the interaction between tubulin and Cx36 which has both tubulin binding motif and CaMKII/CaM binding motif. Owing to loss of tubulin interaction, carboxy-terminal deleted mutant Cx36 impaired electrical plasticity and Cx36 gap junction localization and transport of Neuro-2A cells. By using 3D structural modeling, the authors identified tubulin interface of Cx36 that modulated Cx36 gap junction localization and transport. The approach and the finding are very interesting, but this reviewer has several comments as described below:
Major comments
Figure 2-4. Since the carboxy-terminal (CT) of Cxs might regulate connexon and gap junction formation, the experiment of CT-deleted mutant Cx36 is not enough to investigate impact of tubulin-interaction. The inhibition assay of tubulin-Cx36 interaction by peptide and antibody will support the authors results.
Reply: A TAT-peptide inhibition assay masking the binding motif will be inconclusive since blocking peptides will not discriminate between the ménage trois of proteins, namely tubulin, CaM, and CaMKII, which bind to the same, overlapping protein sequence in the Cx36-CT. Anti-Cx36 antibodies targeting the CT-binding site specifically are presently not available to interfere with tubulin binding. Please note that intracellular perfusion with monomeric tubulin through the recording electrode was equally sufficient to block an increase in junctional currents in comparison to the untreated WT (Fig. 1E,F).
The authors should mention about behavior of CT-deleted mutant Cx43 which has no tubulin binding motif and tubulin motif deleted NMDA receptor or NR2B for comparison with Cx36.
Reply: Thank you for this suggestion. This information is now included in the discussion. Please refer to lines 719 - 730 of the revised manuscript.
Figure 2F and Figure 3, as no references and no appropriate control, this reviewer cannot fully understand the results that CT-deleted mutant Cx36 alters Cx36 localization by loss of interaction with tubulin.
Reply: A wild-type control was used in both experiments. The loss of interaction with tubulin has been previously published (reference 19). The argument that the localization is specific to tubulin is supported by TIRF experiments in which Cx36 containing vesicles were tracked (Figures 3, 6 and S6). Point mutations within the tubulin binding motif refined the analysis.
Figure 5-7, appropriate negative control such as Cx43 is required.
Reply: We argue that Cx43 is not a suitable negative control for Cx36. Cx36 and Cx43 belong to different connexin subclasses. The two proteins differ in length of the CT (and potential interaction motifs), biophysical properties, and expression in functionally distinct cell type. Outcomes using Cx43 in our experimental setups might not be truly “negative.”
It has been thought that altered Cxs expression and function might involve in tumorigenesis and transformation. It is better to describe not only Neuro-2a cells, but also primary cells or other cell lines.
Reply: Cx36 is not known to be involved in tumorigenesis and transformation. Neuro2a cells express very low levels of endogenous Cx36 or other gap junction proteins, which is ideal to study the transport dynamics of transfected Cx36 wild type and mutant proteins. A comparison with primary cells or other cell lines expressing different and potentially higher levels of endogenous Cx36 or other Cx would add confounding factors to this research.
All experiments were performed by in vitro models. The authors need to discuss whether the results in here has physiologically important.
Reply: Good point. The discussion was modified accordingly. Please refer to lines 731-742 of the revised manuscript.
Minor comments
L58, what is wildtype an mutant Cx36? Figure 1, before (-) and after (+) might lead to a misunderstanding. L421, (e) should be replaced to (E).
Reply: Thank you for pointing this out. Corrections were made as appropriate.
Reviewer 2 Report
In the manuscript entitled “Tubulin-dependent transport of connexin-36 potentiates the size and strength of electrical synapses”, by Brown et al., are provided evidence for the existence of a tubulin-interacting domain in Cx36, overlapping a binding site for CaM/CaMKII, that is important for the delivery of Cx36 to GJP. Moreover, the authors claim that tubulin-mediated trafficking of Cx36 to the PM potentiates electrical synapses strength. To address this question, genetic and pharmacological strategies were used, providing a comprehensive view of the subject.
The topic approached in this manuscript is of utmost importance not only in the field of connexins and gap junctions, but also in the context of neuronal function.
The manuscript is very well written and, in most cases, the experiments support the conclusions.
Despite its biological relevance, the manuscript presents some minor flaws that need to be addressed.
Comments and concerns:
. how are the endogenous levels of Cx36 in Neuro-2a cells?
. a decrease in tubulin binding to Cx36 should be provided when using the ∆279-292, when compared with the WT; an IP assay might be performed to show that mutated Cx36 binds less to tubulin in comparison to WT; since the tubulin site overlaps binding domain for CaM/CaMKII, the effect of this Cx36 mutant on the interaction with CaM/CaMKII should be studied.
-it is shown that mutated Cx36 accumulates intracellularly, suggesting an impairment of the delivery to the PM; however, this accumulation may also mirror defects in the internalization process; to clarify this important issue, the following experiments could be performed:
. assess the impact of internalization inhibitors (both CME and NCME) and/ or perform internalization assays
. evaluate the effect of phosphorylation on Cx36 trafficking (it is plausible that some of the effects observed are due to defects of CaMKII-mediated phosphorylation of Cx36)
. unveil whether the degradation of Cx36 is affected in the mutated form (this can be particularly important in the context of ERAD)
. in addition to ER, Golgi and transport vesicles (Caveolin) markers, also endocytic vesicle markers, such as EEA1, should be identified
- to unveil the subcellular distribution of the Cx36 (WT or mutant) in the presence or absence of drugs that affect cytoskeleton-mediated trafficking, biotinylation and subcellular fractionation assays should be performed to assess the relative distribution of Cx36 in the soluble and unsoluble fraction.
- why mutated Cx36 display less colocalization with ER and Golgi than the WT? Is the mutation affecting the oligomerization of Cx36? Is the mutation affecting the ERAD?
- it was previously shown that paclitaxel promotes the transport and stabilization of Cx at the PM; why does this drug impair the binding of Cx36 to ER?
- clarify the impact of paclitaxel on Cx36 transport, either in single cells (only hemichannels are formed, with no GJP) and paired cells (with GJP)
- based on CytoD treatments, it is suggested that actin can serve as a compensatory mechanism (presumably when tubulin-mediated trafficking is impaired); if this is the case, could it be conceivable that mutations (that inhibit interaction with tubulin) increase the interaction with actin and make it more sensible to CytoD
- functional assays (gap junctions and hemichannels) should be considered, namely to what electrical coupling is concerned
- in lane 668 it is stated “gap junction hemichannels”; this should be corrected.
Author Response
Reviewer 2
Comments and Suggestions for Authors
In the manuscript entitled “Tubulin-dependent transport of connexin-36 potentiates the size and strength of electrical synapses”, by Brown et al., are provided evidence for the existence of a tubulin-interacting domain in Cx36, overlapping a binding site for CaM/CaMKII, that is important for the delivery of Cx36 to GJP. Moreover, the authors claim that tubulin-mediated trafficking of Cx36 to the PM potentiates electrical synapses strength. To address this question, genetic and pharmacological strategies were used, providing a comprehensive view of the subject.
The topic approached in this manuscript is of utmost importance not only in the field of connexins and gap junctions, but also in the context of neuronal function.
The manuscript is very well written and, in most cases, the experiments support the conclusions.
Despite its biological relevance, the manuscript presents some minor flaws that need to be addressed.
Comments and concerns:
. how are the endogenous levels of Cx36 in Neuro-2a cells?
Reply: Very low endogenous levels of Cx36 are detectable by RT-qPCR. However, Cx36 protein is not detectable by western blot or immunohistochemistry.
. a decrease in tubulin binding to Cx36 should be provided when using the ∆279-292, when compared with the WT; an IP assay might be performed to show that mutated Cx36 binds less to tubulin in comparison to WT; since the tubulin site overlaps binding domain for CaM/CaMKII, the effect of this Cx36 mutant on the interaction with CaM/CaMKII should be studied.
Reply: BIO-ID was used as IP method in this study (Figure 1A). If required, we can make a conventional IP experiment available. The effects of the Cx36 mutant on CaM binding has been reported previously (Siu et al., FMN 2017; reference # 22). The binding of the ∆272-292 mutant to CaMKII has been reported (del Corrso et al., 2012; reference # 5). Details of the regulation of the Cx36-CaMKII complex including the ∆279-292 at the PM will be reported elsewhere (Siu et al., manuscript in preparation).
-it is shown that mutated Cx36 accumulates intracellularly, suggesting an impairment of the delivery to the PM; however, this accumulation may also mirror defects in the internalization process; to clarify this important issue, the following experiments could be performed:
. assess the impact of internalization inhibitors (both CME and NCME) and/ or perform internalization assays
Reply: In our research, we aim at building a comprehensive mechanistic explanation of how Cx36 is delivered to the PM, and how this enables plasticity of electrical synapses. We share the view that the mechanisms driving Cx36 turnover, removal and the degradation pathways are exciting research objectives. Presently the mechanistic details are beyond the scope of this study.
. unveil whether the degradation of Cx36 is affected in the mutated form (this can be particularly important in the context of ERAD)
Reply: Western blots showed no indication of protein degradation.
. evaluate the effect of phosphorylation on Cx36 trafficking (it is plausible that some of the effects observed are due to defects of CaMKII-mediated phosphorylation of Cx36)
Reply: A very interesting topic, which is part of an ongoing phosphoproteome project we hope to complete within the next 2-3 years.
. in addition to ER, Golgi and transport vesicles (Caveolin) markers, also endocytic vesicle markers, such as EEA1, should be identified
Reply: As mentioned above, the internalization of Cx36 through endocytic vesicle transport, interaction with the microtubule network towards recycling and/or degradation is beyond the focus of this study. However, we agree that this research is of significant interest and merits a separate in-depth study.
- to unveil the subcellular distribution of the Cx36 (WT or mutant) in the presence or absence of drugs that affect cytoskeleton-mediated trafficking, biotinylation and subcellular fractionation assays should be performed to assess the relative distribution of Cx36 in the soluble and unsoluble fraction.
Reply: The imaging of Cx36 particles in living cells offers a superior resolution and insight into the molecular basis of plasticity of GJs. We do not wish to discount the value of classical biochemical approaches, but methods such as subcellular fractionation of membranous compartments are contamination prone and only provide a snapshot of relative distributions of otherwise dynamic transport to membrane and recycling pathways. Our methods aim at providing insight into the dynamic events inside the cell.
- why mutated Cx36 display less colocalization with ER and Golgi than the WT? Is the mutation affecting the oligomerization of Cx36? Is the mutation affecting the ERAD?
Reply: We acknowledged in lines 323-326 that the ER stress response was not addressed in this investigation. As such, we do not have specific knowledge of the mutant’s effect on ERAD. However, protein expression between the mutant and wildtype were equivalent, suggesting that ERAD is not or only marginally involved. A more plausible explanation is the premature release of mutant Cx36 from the ER-Golgi and/or since the protein is still transport competent, a faster mechanism for release is at play. The discussion now includes this as a proposal for less colocalization. Please refer to lines 666-674 of the revised manuscript. This study on tubulin-dependent transport to the PM does not address and provide insight into connexin oligomerization. This process happens before vesicular transport of Cx36 along microtubules is initiated.
- it was previously shown that paclitaxel promotes the transport and stabilization of Cx at the PM; why does this drug impair the binding of Cx36 to ER?
Reply: We apologize for this misunderstanding. We showed its effect on colocalization to the tubulin cytoskeleton (Figure 5E). Impaired binding or co-localization to the ER or Golgi was not demonstrated. For further clarification, we have now included a statement in regards to the lower co-localization observed between Cx36 and tubulin in cells treated with paclitaxel. Please refer to lines 433-436 of the revised manuscript.
- clarify the impact of paclitaxel on Cx36 transport, either in single cells (only hemichannels are formed, with no GJP) and paired cells (with GJP)
Reply: We have not observed substantial expression of unopposed Cx36 hemichannels in our experiments regardless of whether cells were paired or unpaired, which makes such an experiment difficult to perform and interpret.
- based on CytoD treatments, it is suggested that actin can serve as a compensatory mechanism (presumably when tubulin-mediated trafficking is impaired); if this is the case, could it be conceivable that mutations (that inhibit interaction with tubulin) increase the interaction with actin and make it more sensible to CytoD
Reply: Thank you for pointing this out. The discussion was modified accordingly. Please refer to lines 719-742 of the revised manuscript.
- functional assays (gap junctions and hemichannels) should be considered, namely to what electrical coupling is concerned
Reply: We provide functional data on gap junctions in cell pairs treated with cytoskeleton disrupting agents. As mentioned above, we have not found evidence for Cx36 hemichannels in our model, which is consistent with the lack of strong evidence in the Cx36 literature.
- in lane 668 it is stated “gap junction hemichannels”; this should be corrected.
Reply: Thank you for pointing out this error.
Round 2
Reviewer 1 Report
The manuscript has been carefully revised. I also agree with the revision.
Several number of typos has been found. Could the authors check again?
L114, "anti- β-tubulin"
L668, ''wildtype"
L724, "altered reduced"
L727, "pointto"
Reviewer 2 Report
Although the authors addressed some of the issues raised by this reviewer in a vague way, and did not considered the inclusion of new experiments, the changes introduced in the revised version based on my comments have improved the quality of the paper